# A Combinatorial Code for CPEB-Mediated c-myc Repression

**DOI:** 10.3390/cells12192410

**Published:** 2023-10-06

**Authors:** Koichi Ogami, Keima Ogawa, Shoko Sanpei, Fumito Ichikawa, Tsuyoshi Udagawa, Shin-ichi Hoshino

**Affiliations:** 1Department of Biological Chemistry, Graduate School of Pharmaceutical Sciences, Nagoya City University, Nagoya 467-8603, Japanfichikawa@sunfco.com (F.I.); udagawa@phar.nagoya-cu.ac.jp (T.U.); 2Division of Molecular Oncology, Center for Neurological Diseases and Cancer, Nagoya University Graduate School of Medicine, Nagoya 466-8550, Japan

**Keywords:** mRNA deadenylation, CPEB, RNA-binding protein

## Abstract

During early embryonic development, the RNA-binding protein CPEB mediates cytoplasmic polyadenylation and translational activation through a combinatorial code defined by the cy-toplasmic polyadenylation element (CPE) present in maternal mRNAs. However, in non-neuronal somatic cells, CPEB accelerates deadenylation to repress translation of the target, including c-myc mRNA, through an ill-defined cis-regulatory mechanism. Using RNA mutagenesis and electrophoretic mobility shift assays, we demonstrated that a combination of tandemly arranged consensus (cCPE) and non-consensus (ncCPE) cytoplasmic polyadenylation elements (CPEs) constituted a combinatorial code for CPEB-mediated c-myc mRNA decay. CPEB binds to cCPEs with high affinity (Kd = ~250 nM), whereas it binds to ncCPEs with low affinity (Kd > ~900 nM). CPEB binding to a cCPE enhances CPEB binding to the proximal ncCPE. In contrast, while a cCPE did not activate mRNA degradation, an ncCPE was essential for the induction of degradation, and a combination of a cCPE and ncCPEs further promoted degradation. Based on these findings, we propose a model in which the high-affinity binding of CPEB to the cCPE accelerates the binding of the second CPEB to the ncCPEs, resulting in the recruitment of deadenylases, acceleration of deadenylation, and repression of c-myc mRNAs.

## 1. Introduction

Gene expression levels are often determined post-transcriptionally by the act of RNA-binding proteins (RBPs). Various RBPs regulate mRNA translation and stability in the cytoplasm, largely affecting cellular protein levels [1]. Regulation often involves alterations in the poly(A) tail metabolism. Poly(A) tail lengthening, known as polyadenylation, can enhance mRNA translation [2,3,4,5,6], and in contrast, poly(A) tail removal, referred to as deadenylation, causes destabilization of mRNAs [7,8,9]. As a result, RBPs can modulate gene expression both positively and negatively by targeting polyadenylation and deadenylation, respectively.

CPEB is a well-characterized RNA-binding protein that regulates mRNA translation via cytoplasmic polyadenylation [10,11]. CPEB directly binds to U-rich RNA sequences called cytoplasmic polyadenylation elements (CPEs; consensus sequence: UUUUAU and UUUUAAU) present in particular mRNA 3′UTRs. In frog oocytes, CPEB forms a complex with a poly(A) polymerase Gld-2 and polyadenylates its target mRNAs in response to progesterone stimulation [12]. In mammals, CPEB can bind to the poly(A) polymerases PAPD4 (an ortholog of Gld-2, also known as TENT2) and PAPD5 (also known as TENT4B or GLD4) to regulate the poly(A) tail length [3,4].

In addition to CPEB’s role in cytoplasmic polyadenylation, we previously showed that CPEB destabilizes and represses the translation of c-myc mRNA by accelerating deadenylation in human cells [7]. c-myc mRNA was originally identified as a CPEB target in mice, and its regulation is critical for cellular senescence [13]. In human c-myc 3′UTR, there are six potential CPEB-binding motifs: two consensus CPEs (cCPEs) and four non-consensus CPEs (ncCPEs). Introducing point mutations in these motifs completely abrogates CPEB-mediated c-myc mRNA destabilization, indicating that CPEB binds directly to c-myc mRNA and directs its degradation [7]. As a result, CPEB plays a dual role in the regulation of post-transcriptional gene expression by targeting the poly(A) tail metabolism in the opposite directions. However, the mechanism by which CPEB distinguishes mRNA substrates from translationally enhanced or destabilized substrates remains unknown.

In the present study, we identified an RNA sequence encoding CPEB-mediated decay. We prepared various c-myc reporters with 3′UTR mutations and evaluated the effect of CPEB on the level of reporter mRNAs as well as CPEB-binding affinities. Our findings indicated that ncCPEs are indispensable for c-myc mRNA decay, but CPEB exhibits a low affinity for them. Detailed in vitro RNA-protein-binding assays revealed that a CPEB bound to a cCPE can enhance a second CPEB binding to an ncCPE. This study highlights the importance of the composition of CPEB binding elements in c-myc mRNA decay.

## 2. Materials and Methods

### 2.1. Plasmids

The construction of pHA-CPEB, pCMV-5×myc-CPEB, pCMV-5×Flag-CPEB, pMAL-cRI-CPEB, and pFlag-CMV5/TO-BGG c-myc 3′UTR was described previously [7]. HA-tagged CPEB was amplified from the pHA-CPEB vector using the primer pairs HA-hCPEB-F and HA-hCPEB-R (listed in Appendix A) and cloned into the pCDH-EF1 lentiviral vector at the NheI site to generate pCDH-EF1-HA-hCPEB1. c-jun and p53 3′UTR sequences were amplified using HeLa genomic DNAs. All c-myc 3′UTR mutants were generated by inverse PCR using the primers and templates listed in Appendix A.

### 2.2. Northern Blotting and Western Blotting

Northern and Western blotting analyses were performed as described previously [7].

### 2.3. Cell Culture and Transfection

HeLa cells were cultured in Dulbecco’s modified Eagle’s medium (DMEM; Nissui Pharmaceutical, Tokyo, Japan) supplemented with 5% fetal bovine serum. Transfection of plasmid DNA was performed using Lipofectamine 2000 (Thermo Fisher Scientific, Waltham, USA) as previously described [7].

### 2.4. Lentivirus Production

Lentiviruses were produced using the lentiviral vectors with the packaging vectors pSPAX2 and pMD2.G in LentiX-293T cells (TaKaRa, Kusatsu, Japan). Transfection was performed in 6 cm dishes using the PEI-Max reagent (Polysciences, Warrington, PA, USA) according to the manufacturer’s instructions. Four hours after transfection, the medium was replaced with fresh DMEM containing 10% FBS, and the virus-containing medium was collected and filtered two days later.

### 2.5. RNA Electrophoretic Mobility Shift Assay (REMSA)

For REMSA, wild-type or mutant c-myc 3′UTR RNAs were transcribed from PCR-amplified DNA templates using T7 RNA polymerase (TaKaRa, Kusatsu, Japan) in the presence of ^32^P-UTP. A 15 fmol portion of a ^32^P-labeled, gel-purified RNA probe was denatured at 65 °C for 10 min and cooled to room temperature. The refolded RNA was incubated with MBP fusion proteins in an RNP binding buffer (10 mM HEPES-KOH pH 7.9, 50 mM KCl, 1 mM MgCl_2_, 0.1 mM ZnCl_2_, 0.1% NP-40, 1 mM DTT, 2 μg yeast tRNA, 2 μg BSA, and 12 U RNaseOUT) at a final volume of 10 μL. The binding reaction mixture was then incubated at room temperature for 40 min. Heparin (50 ng) was added. The binding reaction (5 μL) was run in a 5% non-denaturing polyacrylamide gel in a 0.5 × TBE buffer. The gels were dried on a filter using a RAPIDDRY-MINI (ATTO, Tokyo, Japan AE-3711), and the bands were detected by autoradiography. Band intensities were estimated using MultiGauge software (Fujifilm, Tokyo, Japan).

## 3. Results

### 3.1. c-myc mRNA Is Specifically Degraded by CPEB

We have previously reported that CPEB promotes the deadenylation and decay of c-myc mRNA [7]; it is well characterized as a regulator of cytoplasmic polyadenylation. As a result, it is unlikely that CPEB promotes targeting by recognizing the same cis-regulatory elements as the mRNAs subjected to cytoplasmic polyadenylation. To evaluate the specificity of CPEB-mediated mRNA decay, we examined the effects of CPEB overexpression on reporter mRNAs harboring a different 3′UTR from known CPEB targets (c-myc, p53, and c-jun) [3,4,14]. While the exogenous expression of CPEB resulted in a decrease in reporter mRNAs with the c-myc 3′UTR (Figure 1), there was no significant change in mRNA levels with p53 or the c-jun 3′UTR. These findings indicate that the presence of cCPE alone cannot direct CPEB to promote mRNA degradation, and that the c-myc 3′UTR contains a specialized combination of *cis*-regulatory elements.

### 3.2. ncCPE Sequences Are Indispensable for CPEB-Mediated c-myc mRNA Decay

To determine the sequence elements of the c-myc 3′UTR required for CPEB-mediated decay, we first constructed two reporter genes appended with c-myc 3′UTR deletion mutants (pA1 and pA2; Figure 2a,b). The pA1 reporter contained the first half of the c-myc 3′UTR sequence, and the pA2 contained the rest of the sequence. Northern blotting revealed that both of the reporter mRNAs were degraded by CPEB expression to the same extent as the wild-type reporter, implying that the responsible elements were present in both regions (Figure 2a,b). To further narrow down the sequence, we prepared various deletion mutants of the pA2 reporter (Figure 2d,e). As the sequence narrowed, the extent of degradation by CPEB expression decreased. Reporters that lacked the cCPE were partially affected but still sensitive to CPEB expression (5′Δ2, 5′Δ3). In contrast, the 3′Δ2 reporter that contained the cCPE but lacked both of the ncCPEs was found to be resistant to CPEB-mediated decay.

To confirm the importance of the ncCPEs, we introduced various combinations of point mutations into the pA2 reporter and examined their sensitivity to CPEB expression (Figure 3a,b). Again, a single-point mutation into the consensus CPE only partially affected CPEB sensitivity (CPE mt). In sharp contrast, and most importantly, mutations in both ncCPEs completely abrogated the CPEB-mediated degradation of mRNA (ncCPE1/2 mt). Consistent with the findings of deletion mutants, the presence of either of the ncCPEs maintained CPEB sensitivity (5′Δ3 and 3′Δ1 in Figure 2d,e and CPE/ncCPE1 mt and CPE/ncCPE2 mt in Figure 3a,b).

Point mutations in the pA1 reporter further demonstrated the importance of ncCPE sequences in CPEB-mediated decay (Figure 4a,b). Then, we investigated whether CPEB accelerates mRNA decay through the c-myc 3′UTR in normal diploid IMR-90 cells. As shown in Appendix A, CPEB indeed accelerates the decay of the reporter mRNA in IMR-90 cells. Furthermore, mutations in both ncCPEs in pA2 completely abolished CPEB-mediated degradation of the reporter mRNA, while a single-point mutation in the cCPE had minimal effect. These results strongly suggest that ncCPEs are essential for CPEB-mediated c-myc mRNA decay in both cell types. In addition, the findings imply that the cCPE is dispensable but still participates in decay.

### 3.3. sCPE Helps a Second CPEB Bind to ncCPE Sequences

Despite the importance of ncCPEs in CPEB-mediated c-myc mRNA decay, it remains unclear whether CPEB efficiently binds to these sequences. To compare the binding affinities of CPEB to the cCPE and ncCPE sequences of the c-myc 3′UTR pA2 region, REMSA (RNA electrophoretic mobility shift assay) was performed using radiolabeled c-myc 3′UTR pA2 RNA probes and a recombinant MBP-CPEB protein. To estimate the binding affinity of each possible CPEB-binding sequence, we prepared various combinations of c-myc 3′UTR mutant transcripts. As expected, CPEB showed a higher affinity for CPEs (Figure 5a, estimated Kd of 212 nM) than for ncCPEs (Figure 5b,c; estimated Kd of 890 nM for ncCPE1 and > 1000 nM for ncCPE2). No significant CPEB binding was observed even at the highest protein concentration when all possible CPEB binding sites were mutated, indicating that CPEB was capable of binding to ncCPEs (Figure 5d).

Intriguingly, when using a wild-type RNA probe, the binding of two CPEB molecules was observed (Figure 6a, indicated by an asterisk, with an estimated Kd of 279 nM). The shift that represents single CPEB binding appears at a lower concentration, equivalent to that of the cCPE RNA probe, suggesting that the faster migrating shift band represents binding to the cCPE (compare Figure 6a and Figure 5a). The fact that the second CPEB binding was only observed in REMSA using an RNA probe with both the cCPE and ncCPE implies that the second CPEB binding requires both the ncCPE and cCPE. Consistent with this idea, a second CPEB binding was not observed with the RNA probe with two ncCPEs and a mutated cCPE, which showed a low affinity for CPEB (compare Figure 6b and Figure 5b,c, estimated Kd for ncCPE1/2 of 891 nM). Notably, the second binding was observed at a much lower protein concentration in the wild-type RNA probe than expected from its affinity for ncCPEs (compare Figure 6a and Figure 6b). Ultimately, we aimed to clarify the ncCPE sequence responsible for the second CPEB binding. REMSA experiments using RNA probes harboring a cCPE/ncCPE1 or cCPE/ncCPE2 pair showed that the second CPEB binding could be observed with both probes, although a pair of cCPE and ncCPE1 could more efficiently recruit a second CPEB molecule (Figure 6c,d). Overall, these results suggest that a CPEB binding to the cCPE facilitates the second CPEB binding to the ncCPEs. This mechanism enables CPEB to bind to ncCPEs even at low CPEB concentrations.

## 4. Discussion

The understanding of the RBP-binding sequences and the functions and composition of RBP-containing protein complexes has progressed remarkably; however, knowledge regarding the combinatorial effects of *cis*-regulatory elements remains limited. In this study, we found that a combination of tandemly arranged cCPE and ncCPE elements was involved in CPEB-mediated c-myc mRNA decay. In contrast to the combinatorial code demonstrated for CPEB-mediated cytoplasmic polyadenylation, the cCPE alone did not induce CPEB-mediated degradation of the c-myc reporter mRNA. CPEB bound to an ncCPE is responsible for the onset of c-myc mRNA decay. Consistent with this idea, when an ncCPE was inserted into the 3′UTR of p53 mRNA, the artificial cCPE-ncCPE chimeric code accelerated the degradation of p53 mRNA (Appendix A). Additionally, despite the requirement of an ncCPE for the decay, CPEB exhibited a much lower binding affinity toward ncCPEs than cCPEs. However, when cCPEs coexisted on the same RNA probe, the second CPEB binding to an ncCPE was promoted by the first CPEB binding to a cCPE with high affinity. These findings indicated that the cCPE plays a role in supporting the recruitment of another CPEB to a low-affinity ncCPE element. Consistent with this notion, the insertion of the hGH sequence between the cCPE and ncCPE of c-myc mRNA abrogated the degradation of c-myc mRNA (Appendix A). Further studies will determine the extent to which these findings are universally applicable to any cell type and species. Currently, the underlying biochemical mechanisms explaining how cCPE- and ncCPE-bound CPEBs play distinct roles in c-myc mRNA decay remain unclear. There might be conformational differences between cCPE- and ncCPE-bound CPEBs, or, alternatively, the c-myc 3′UTR secondary structure may be rearranged by the first CPEB binding to a cCPE. We favor the former explanation because we have previously shown that the artificial MS2 tethering of CPEB to the 3′UTR can efficiently promote mRNA decay [7]. CPEB contains two RNA recognition motifs (RRMs) in its C-terminal region, and the orientation of these RRMs changes when bound to RNA [15]. Our results using p53 and c-jun reporters, both of which contain a cCPE but lack an ncCPE, as well as mutated c-myc 3′UTR, suggest that CPEB cannot promote mRNA decay when tightly bound to consensus sequences. In contrast, when CPEB binds to mRNA less tightly via ncCPE or RNA motifs independently through the MS2-tethering system, it induces mRNA decay. The RNA-binding state of CPEB RRMs may determine whether the mRNA is polyadenylated or deadenylated. The latter RNA secondary structure hypothesis is also possible, and it is reminiscent of the combinatorial effect of CPEB and Msi1 in the regulation of c-mos mRNA, in which Msi1 binding to the Musashi-binding element unfolds the c-mos 3′UTR and reveals the CPE available for CPEB binding required for cytoplasmic polyadenylation [16]. Further research is needed to investigate the detailed mechanism of the second CPEB-binding promotion to an ncCPE by CPEB bound to a cCPE as well as to determine whether similar combinatorial codes are employed in other mRNAs.

## 5. Conclusions

In summary, we identified a characteristic RNA sequence code for CPEB-mediated c-myc mRNA decay, which consists of consensus and non-consensus CPEs and provides novel insight into how RBP may regulate mRNA metabolisms depending on the contexture of the RNA sequence motifs.

## Figures and Tables

**Figure 1 cells-12-02410-f001:**
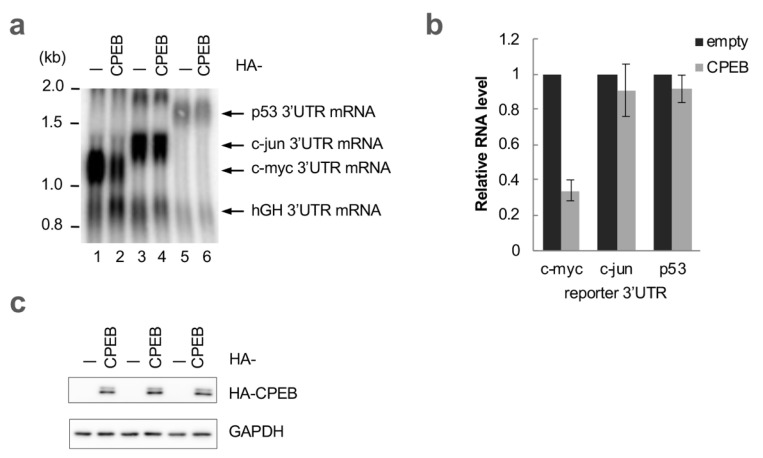
CPEB specifically destabilizes mRNA with the c-myc 3′UTR. (**a**) HeLa cells were transfected with pFlag-CMV5/TO-BGG c-myc 3′UTR, c-jun 3′UTR or p53 3′UTR reporter plasmids, pFlag-CMV5/TO-BGG hGH 3′UTR reference plasmid, and either pHA-CMV5-CPEB or pHA-CMV5. Total RNA was prepared from the cells and subjected to Northern blotting. (**b**) Relative BGG mRNA levels in CPEB-overexpressed cells as in (**a**) were estimated and normalized by a reference hGH 3′UTR reporter mRNA level. The data are presented as the mean ± standard deviation (SD) (*n* = 3). (**c**) Western blotting showing the expression of HA-CPEB as in (**a**). HA-CPEB and GAPDH were used as loading controls.

**Figure 2 cells-12-02410-f002:**
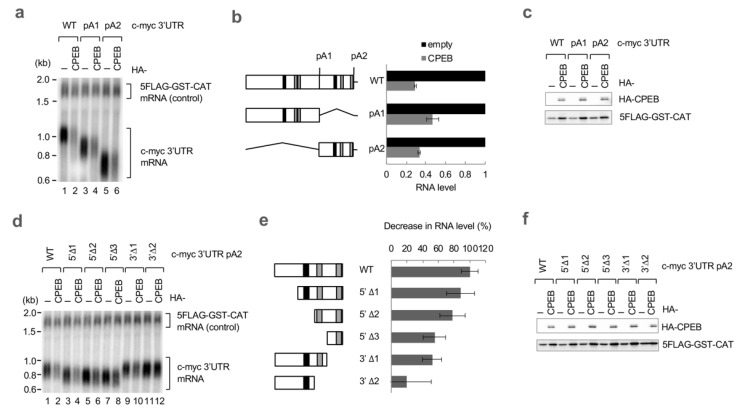
Both cCPE and ncCPE sequences direct CPEB-mediated c-myc mRNA decay. (**a**) HeLa cells were transfected with pFlag-CMV5/TO-BGG c-myc 3′UTR or its mutant reporter plasmid, pCMV-5×Flag-GST-CAT reference plasmid, and either pHA-CMV5-CPEB or pHA-CMV5. Total RNA was prepared from the cells and subjected to Northern blotting. (**b**) Schematic diagram of the c-myc 3′UTR and its deletion mutants (left). Wild-type c-myc 3′UTR contains two consensus CPEs (cCPEs; black box), four non-consensus CPEs (ncCPEs; gray box), and two polyadenylation sites (pA1, pA2). Relative BGG mRNA levels in CPEB-overexpressed cells, as in (**a**), were estimated and normalized by GST-CAT mRNA (right). The data are presented as the mean ± SD (*n* = 3). (**c**) Western blotting showing the expression of HA-CPEB as in (**a**). 5FLAG-GST-CAT was served as a transfection control. (**d**) HeLa cells were transfected with pFlag-CMV5/TO-BGG c-myc 3′UTR pA2 or its deletion mutant reporter plasmid, pCMV-5×Flag-GST-CAT reference plasmid, and either pHA-CMV5-CPEB or pHA-CMV5. Total RNA was prepared from the cells and subjected to Northern blotting. (**e**) Schematic diagram of c-myc 3′UTR pA2 deletion mutants (left). The cCPEs and ncCPEs are depicted as black and gray boxes, respectively. Relative BGG mRNA levels in CPEB-overexpressed cells as in (**c**) were estimated and normalized by GST-CAT mRNA (right). The data are presented as the mean ± SD (*n* = 3). (**f**) Western blotting showing the expression of HA-CPEB as in (**d**). 5FLAG-GST-CAT was used as a transfection control.

**Figure 3 cells-12-02410-f003:**
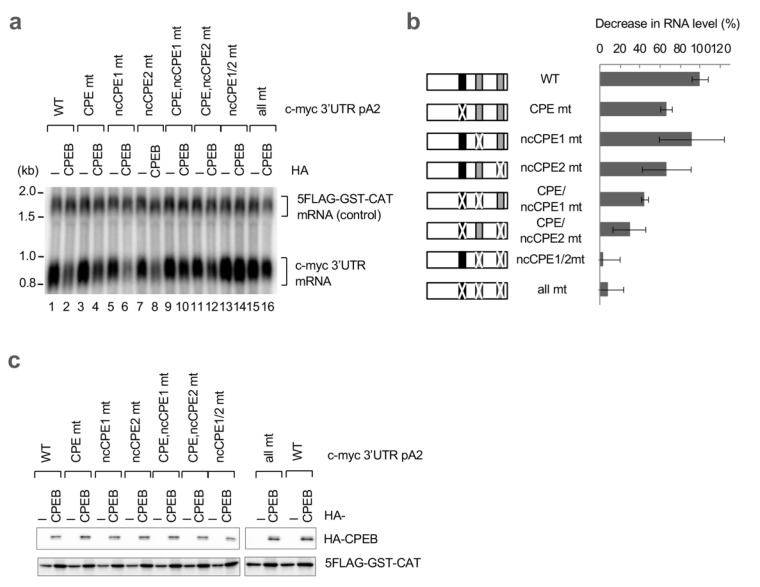
ncCPEs in the c-myc 3′UTR are indispensable for CPEB-mediated mRNA decay. (**a**) HeLa cells were transfected with pFlag-CMV5/TO-BGG c-myc 3′UTR pA2 or its point mutation reporter plasmid, pCMV-5×Flag-GST-CAT reference plasmid, and either pHA-CMV5-CPEB or pHA-CMV5. Total RNA was prepared from the cells and subjected to Northern blotting. (**b**) Schematic diagram of c-myc 3′UTR pA2 mutants (left). The cCPEs and ncCPEs are depicted as black and gray boxes, respectively. The crossbars on the boxes represent the mutated motifs. Relative BGG mRNA levels in CPEB-overexpressed cells, as in (**a**), were estimated and normalized by GST-CAT mRNA (right). The data are presented as the mean ± SD (*n* = 3). (**c**) Western blotting showing the expression of HA-CPEB, as in (**a**). 5FLAG-GST-CAT was used as a transfection control.

**Figure 4 cells-12-02410-f004:**
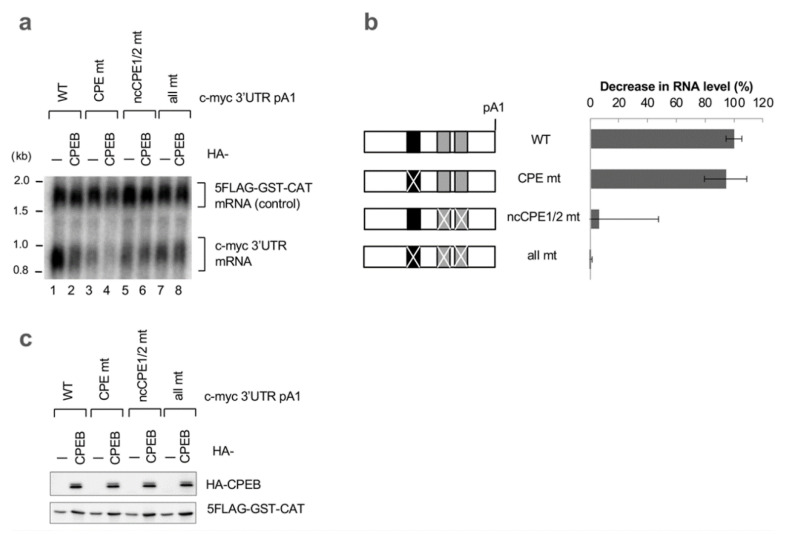
Alternatively, polyadenylated c-myc 3′UTR are degraded by CPEB in a ncCPE-dependent manner. (**a**) HeLa cells were transfected with pFlag-CMV5/TO-BGG c-myc 3′UTR pA1 or its point mutation reporter plasmid, pCMV-5×Flag-GST-CAT reference plasmid, and either pHA-CMV5-CPEB or pHA-CMV5. Total RNA was prepared from the cells and subjected to Northern blotting. (**b**) Schematic diagram of c-myc 3′UTR pA1 mutants (left). The cCPEs and ncCPEs are depicted as black and gray boxes, respectively. The crossbars on the boxes represent the mutated motifs. Relative BGG mRNA levels in CPEB-overexpressed cells, as in (**a**), were estimated and normalized by GST-CAT mRNA (right). The data are presented as the mean ± SD (*n* = 3). (**c**) Western blotting showing the expression of HA-CPEB as in (**a**). 5FLAG-GST-CAT was used as a transfection control.

**Figure 5 cells-12-02410-f005:**
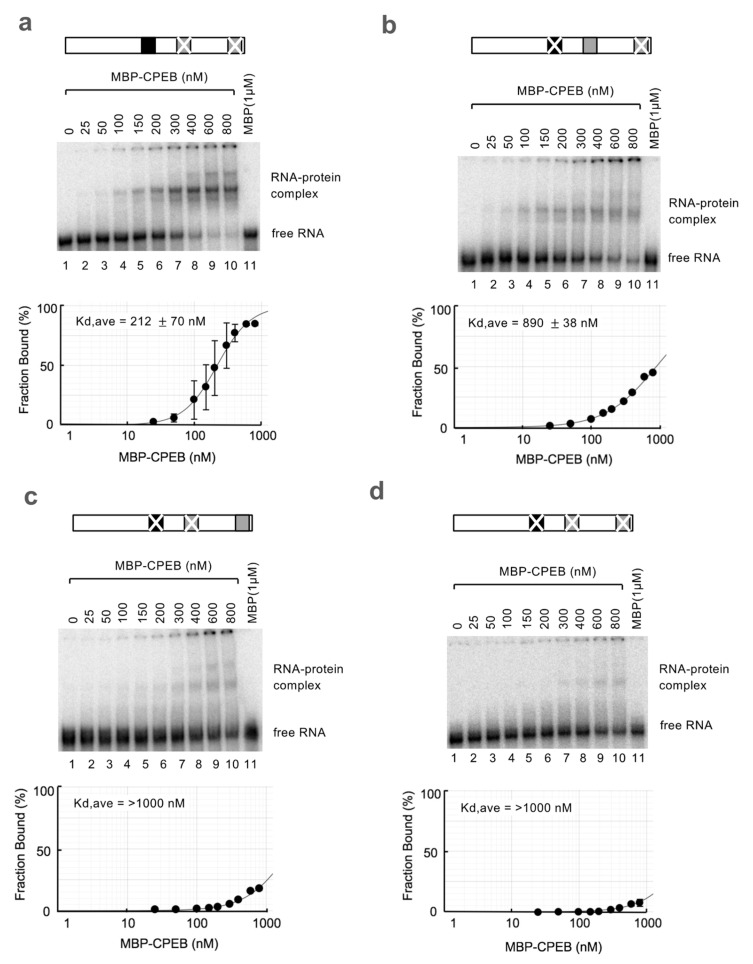
The affinity of CPEB for cCPEs is much higher than for ncCPEs. (**a**–**d**) REMSA assays using in vitro transcribed c-myc pA2 RNA probes and purified recombinant CPEB protein. Schematic diagrams of the RNA probes are shown at the *top*. The cCPEs and ncCPEs are depicted as black and gray boxes, respectively, and the crossbars on the boxes represent the mutated motifs. The representative gel images are shown in the *middle*, and the results of the estimated fraction bound are shown at the *bottom*. The experiments were repeated at least three times, and the average Kd values with SD are shown.

**Figure 6 cells-12-02410-f006:**
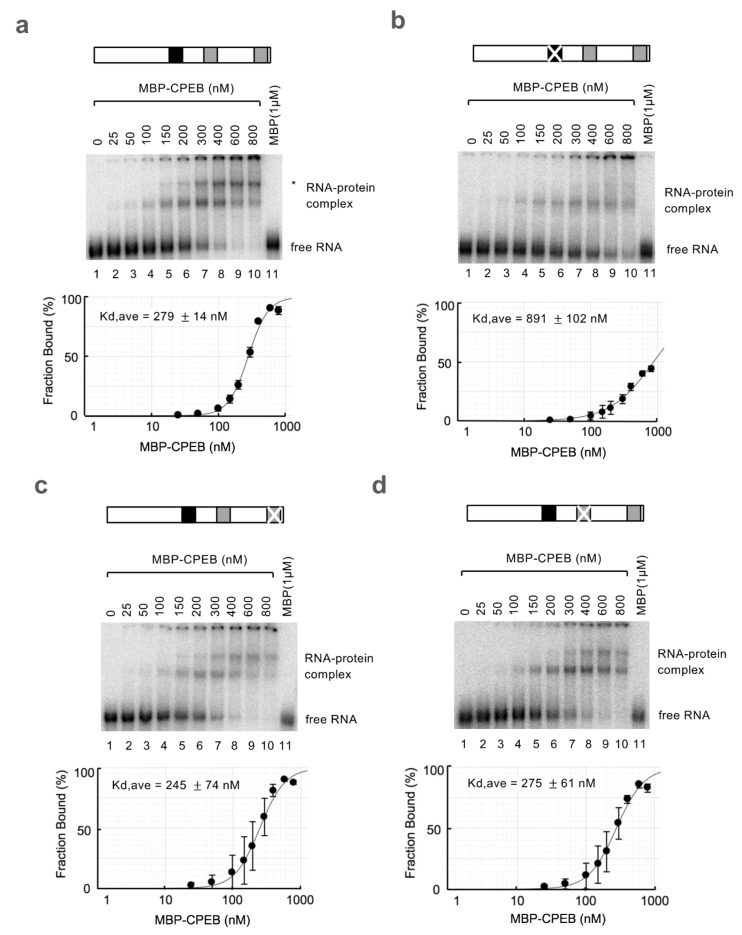
CPEB bound to cCPE helps another CPEB binding to an ncCPE. (**a**–**d**) REMSA assays using in vitro transcribed c-myc pA2 RNA probes and purified recombinant CPEB protein. Schematic diagrams of the RNA probes are shown at the *top*. The cCPEs and ncCPEs are depicted as black and gray boxes, respectively, and the crossbars on the boxes represent the mutated motifs. The representative gel images are shown in the *middle*, and the results of the estimated fraction bound are shown at the *bottom*. The experiments were repeated at least three times, and the average Kd values with SD are shown.

## Data Availability

All data are available in the main text or the Appendix A.

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
