# Peer review of "A Combinatorial Code for CPEB-Mediated c-myc Repression"

_cells, 2023, doi:10.3390/cells12192410_

Round 1

Reviewer 1 Report (Previous Reviewer 2)

I am satisfied with the authors' revision.

Author Response

We are very pleased to learn that Reviewer 1 is now satisfied with our manuscript.  We thank the reviewer for providing so far the constructive comments, valuable suggestions to strengthen our conclusions and positive evaluation of our work.

Reviewer 2 Report (New Reviewer)

This is a very interesting initial study. Ogami et al. show that CPEB is capable of causing mRNA decay of Myc (and in control experiments of other genes, including p53) when cCPE and ncCPE sites are present. cCPE alone is not sufficient to initiate mRNA decay. The study is performed in HeLa cells.

Evaluation of the manuscript.

1.     The manuscript is clearly written (albeit some English corrections are highly recommended).

2.     The findings appear solid for HeLa cells. However, the question arises whether the same data would be observed in primary cells (primary fibroblasts at early passage or primary lymphocytes). The immortalization and transformation process are known to change regulatory processes. The reader needs to be assured that this paper will demonstrate findings that are relevant to HeLa alone or to normal cells as well. If in fact, the findings apply to HeLa alone, the question arises whether other tumor cell lines as well as primary tumors show the same regulatory cCPE/ncCPE mechanisms. Until this is understood, the data do not allow the reader to assess the general relevance of the findings.

3.     The title “A combinatorial code…” is too strong (see comments under 2). It implies that this combinatorial code is required in general for Myc repression when the data are derived from HeLa cells alone.

Minor comments.

1.     Some text is in red. Why?

2.     The English should be double-checked by a native speaker as there are some grammatical errors.

Please improve the language.

Author Response

Response to Reviewer 2 Comments

This is a very interesting initial study. Ogami et al. show that CPEB is capable of causing mRNA decay of Myc (and in control experiments of other genes, including p53) when cCPE and ncCPE sites are present. cCPE alone is not sufficient to initiate mRNA decay. The study is performed in HeLa cells.

--- We thank the reviewer for the positive evaluation of our work.

 Evaluation of the manuscript.

  1. The manuscript is clearly written (albeit some English corrections are highly recommended).

--- Thank you for the comment. As recommended by the reviewer, we submitted our manuscript to English editing services and made corrections on grammatical errors.

  1. The findings appear solid for HeLa cells. However, the question arises whether the same data would be observed in primary cells (primary fibroblasts at early passage or primary lymphocytes). The immortalization and transformation process are known to change regulatory processes. The reader needs to be assured that this paper will demonstrate findings that are relevant to HeLa alone or to normal cells as well. If in fact, the findings apply to HeLa alone, the question arises whether other tumor cell lines as well as primary tumors show the same regulatory cCPE/ncCPE mechanisms. Until this is understood, the data do not allow the reader to assess the general relevance of the findings.

--- We appreciate the reviewer’s comments on the relevance of our findings. Our current study focused on the regulatory mechanisms and c-myc mRNA decay pathway in HeLa cells, specifically through gain-of-function experiments with overexpressed CPEB. However, we acknowledge that our findings in HeLa cells, combined with previous evidence in other cell lines, suggest a general applicability of the mechanisms. In our previous paper (Oncogene Ogami et al., 2014), we demonstrated that the stabilization of c-myc mRNA upon CPEB knockdown in U2OS cells, confirming the involvement of CPEB in the regulation of c-myc mRNA decay at least in the two different cell lines. In the paper, we also showed that CPEB is absent in HeLa cells. In this regard, we revealed the contribution of CPEB in c-myc mRNA stability by loss-of-function (knockdown in U2OS cells) and gain-of-function (moderate overexpression in HeLa cells) experiments. This highlights the broader role of CPEB in mRNA regulation beyond HeLa cells. Although we have not performed experiments using primary cells, our current and previous evidence strongly suggest that the cCPE/ncCPE-supported c-myc mRNA decay mechanism should operate in cells expressing all subunits of the CPEB-Tob-Caf1 complex and c-myc mRNA. We should note that we do not mean that this mechanism is applicable to all tissues and cell types. In some specific cases, such as cell transformation and immortalization, the CPEB-mediated c-myc decay system may be perturbed through altered expression of the responsible factors, activation/repression of signaling pathways. Although Tob and Caf1 are ubiquitously expressed in normal tissues (Matsuda et al., Oncogene 1996; Chen et al., BBRC 2011), CPEB may be absent in some specific cell types like HeLa cells. Pursuing de-regulation mechanisms as well as c-myc mRNA level regulation in CPEB-absent cells would be interesting theme for future work.

  1. The title “A combinatorial code…” is too strong (see comments under 2). It implies that this combinatorial code is required in general for Myc repression when the data are derived from HeLa cells alone.

--- As mentioned above (response to the comment 2), we showed in our previous study that the contribution of CPEB in c-myc mRNA stability by loss-of-function (knockdown in U2OS cells) and by gain-of-function (moderate overexpression in HeLa cells) experiments, suggesting that the cCPE/ncCPE-supported c-myc mRNA decay mechanism described in this study should broadly operate in cells expressing all subunits of the CPEB-Tob-Caf1 complex and c-myc mRNA. Given these factors, we cordially ask your understanding in maintaining the current title.

Minor comments.

  1. Some text is in red. Why?

--- The manuscript we colored the changes in red is the 2nd revised version of our manuscript. According to the reviewers’ suggestions, we have performed new experiments over the course of 6 months.  Especially, we have added new data showing that (i) ncCPE insertion into the 3’UTR of p53 mRNA causes accelerated degradation of the p53 mRNA (Supplementary Figure1 a-d), and (ii) insertion of hGH sequence between cCPE and ncCPE of c-Myc mRNA causes stabilization of the c-Myc mRNA (Supplementary Figure1 e-g), whose results are consistent with and strengthen our conclusion that CPEB mediates c-Myc mRNA decay through binding to the combinatorial code consisting of cCPE and ncCPEs. We expected that the revised manuscript was evaluated by the two original reviewers. Although the reviewer 1 took over and was satisfied with our revised manuscript at this round, the original reviewer 2 seems to be changed for some reason.

  1. The English should be double-checked by a native speaker as there are some grammatical errors.

--- We submitted our manuscript to English editing services and made corrections on grammatical errors.

Round 2

Reviewer 2 Report (New Reviewer)

The authors have responded to my comments. I would like to further elaborate on the use of HeLa cells only (this study) and the authors' general conclusions for other cell lines.

The cited paper (Ogami 2014) does not describe the same findings related to CPEB.  No studies similar to the one submitted here for review have been done in normal (primary) cells.

Thus, the general conclusion must be rephrased for HeLa cells only discussing the potential relevance for other tumor cell lines and possibly normal cells. 

In addition, the title of the paper should have the addition of "...in HeLa cells". I would like the information given to the reader to be exact.

Science is exact and should not include over interpretation. The generalization of the findings is premature based on other data available to date.

Author Response

The authors have responded to my comments. I would like to further elaborate on the use of HeLa cells only (this study) and the authors' general conclusions for other cell lines.

The cited paper (Ogami 2014) does not describe the same findings related to CPEB.  No studies similar to the one submitted here for review have been done in normal (primary) cells.

Thus, the general conclusion must be rephrased for HeLa cells only discussing the potential relevance for other tumor cell lines and possibly normal cells. 

In addition, the title of the paper should have the addition of "...in HeLa cells". I would like the information given to the reader to be exact.

Science is exact and should not include over interpretation. The generalization of the findings is premature based on other data available to date.

--- Thank you for your comment. 

According to the reviewer 2’s comment, we used primary normal diploid IMR-90 cells to investigate whether CPEB accelerates mRNA decay through the c-myc 3’UTR, similar to its effects in HeLa cells.  As shown in Supplementary Figure 1, CPEB indeed accelerates the decay of the reporter mRNA in IMR-90 cells.  Furthermore, mutations in both non-consensus CPEs in pA2 completely abolished CPEB-mediated degradation of the reporter mRNA, while a single point mutation in the consensus CPE had minimal effect.  These results align with the data obtained in HeLa cells, suggesting that CPEB accelerates mRNA decay primarily through non-consensus CPEs, even in primary normal diploid IMR-90 cells.  These findings have been incorporated as a new Supplementary Figure 1.

This manuscript is a resubmission of an earlier submission. The following is a list of the peer review reports and author responses from that submission.

Round 1

Reviewer 1 Report

It is well-known that CPEB enhances mRNA translation through cytoplasmic polyadenylation. In addition to its positive function, the authors have previously shown that CPEB accelerates deadenylation of c-myc mRNA. In the manuscript by Omagi et al., the authors present fine and interesting data describing a combinatorial code for c-myc mRNA decay mediated by ncCPE and cCPE. Noteworthy is that ncCPE is essential for the promotion of c-myc mRNA degradation, while cCPE alone fails to induce the decay.

The manuscript is well-written. Overall the results are clear and convincing to support their claims of a combinatorial code for CPEB-mediated c-myc mRNA degradation. This study greatly contributes to the understanding of the mechanism of a dual, opposite function of CPEB in the poly(A) tail metabolism. I have only a few comments that may be helpful for improvement of the manuscript.

 General comments

1. Please show the immunoblot images to demonstrate the (equal) expression of HA-CPEB (Figs. 1-4), if possible. It may be important especially in Figs. 2a, 3, and 4., because previous study by the authors showed that negative effect of CPEB on BGG c-myc reporter is dose-dependent (Ogami et al., Oncogene, 2014).

2.  The authors previously reported that CPEB-induced degradation of c-myc mRNA is initiated by deadenylation. Although that is true, there is no experimental evidence showing deadenylation in this study. I’d like to suggest the authors distinguish between “deadenylation” and “degradation (decay)” in the text.

Specific comments

1.  Lines 98-99: Does it mean that p53 and c-jun 3’-UTRs contain only consensus CPE? Indeed, p53 mRNA undergoes CPEB-dependent cytoplasmic polyadenylation (Burns and Richter, Genes Dev., 2008).

2.   Lines 228-230: It seems that there is no predicate, only subject clause. “remains unclear” follows?

3.      Line 117: According to Fig. 2D, 3’Δ1 still contains the consensus CPE.

4.      Line 50: It is recommended to cite Ref. [7] after “to be degraded”.

5.      Line 154: It may be better to put a word “further” in front of “showed”.

Author Response

Response to reviewer's comments

It is well-known that CPEB enhances mRNA translation through cytoplasmic polyadenylation. In addition to its positive function, the authors have previously shown that CPEB accelerates deadenylation of c-myc mRNA. In the manuscript by Omagi et al., the authors present fine and interesting data describing a combinatorial code for c-myc mRNA decay mediated by ncCPE and cCPE. Noteworthy is that ncCPE is essential for the promotion of c-myc mRNA degradation, while cCPE alone fails to induce the decay.

The manuscript is well-written. Overall the results are clear and convincing to support their claims of a combinatorial code for CPEB-mediated c-myc mRNA degradation. This study greatly contributes to the understanding of the mechanism of a dual, opposite function of CPEB in the poly(A) tail metabolism. I have only a few comments that may be helpful for improvement of the manuscript.

 General comments

Point 1: Please show the immunoblot images to demonstrate the (equal) expression of HA-CPEB (Figs. 1-4), if possible. It may be important especially in Figs. 2a, 3, and 4., because previous study by the authors showed that negative effect of CPEB on BGG c-myc reporter is dose-dependent (Ogami et al., Oncogene, 2014).

Response 1: We agree with the reviewer’s point. We have added western blotting images showing HA-CPEB expression (Figures 1c, 2c, 2f, 3c and 4c).

Point 2:  The authors previously reported that CPEB-induced degradation of c-myc mRNA is initiated by deadenylation. Although that is true, there is no experimental evidence showing deadenylation in this study. I’d like to suggest the authors distinguish between “deadenylation” and “degradation (decay)” in the text.

Response 2: We used “degradation” or “decay” instead of “deadenylation” when describing the evidence obtained in this study.

Specific comments

Point 1:   Lines 98-99: Does it mean that p53 and c-jun 3’-UTRs contain only consensus CPE? Indeed, p53 mRNA undergoes CPEB-dependent cytoplasmic polyadenylation (Burns and Richter, Genes Dev., 2008).

Response 1: P53 and c-jun 3’UTRs contain consensus CPE but lacks non-consensus CPE-like sequences. We have clearly mentioned this in the discussion (lines 263 – 264).

Point 2:   Lines 228-230: It seems that there is no predicate, only subject clause. “remains unclear” follows?

Response 2: We have added “remains unclear” at the end of the sentence.

Point 3:      Line 117: According to Fig. 2D, 3’Δ1 still contains the consensus CPE.

Response 3: We erroneously included 3’Δ1. We have corrected the sentence.

Point 4:      Line 50: It is recommended to cite Ref. [7] after “to be degraded”.

Response 4: We have added Ref. [7] in the sentence.

Point 5:      Line 154: It may be better to put a word “further” in front of “showed”.

Response 5: We have put a word “further” accordingly.

Reviewer 2 Report

In the manuscript by Ogami et al., they revealed that CPEB overexpression induced degradation of c-Myc mRNA. In addition, ncCPEs within the 3’ UTR of c-myc mRNA contribute to the degradation. They propose that a CPEB which binds to a cCPE recruit another CPEB molecule to an ncCPE, which is indispensable for degradation. The quality of the data is quite good and writing is nice. However, some additional experiments need to be done to fully support their conclusion. Here are my comments: 

Major comments

1.     The author argue that CPEBs binding to ncCPEs contribute to mRNA degradation. If so, ncCPE insertion to the 3’ UTR of p53 and c-jun should cause degradation of them.  

2.     If the binding is important for the degradation, the binding efficiency should be directly proportional to the degradation rate. However, in Figure 3b, the degradation efficiencies of CPE mt and ncCPE2 mt are almost the same, however the binding efficiency (Kd) of ncCPE2 was much lower than CPE mt (Figure 6). Why so different? In addition, if a CPEB binding to an cCPE recruits another CPEB to a ncCPE, insertion of additional sequence between cCPE and ncCPE might inhibit the recruitment of CPEB to ncCPE and degradation of mRNA. 

3.     In Figure 4a, c-myc mRNA band of CPE mt is weaker than the others. Does this cCPE contribute to polyadenylation and stabilization of the mRNA? 

Minor comments

1.     Put a space between the numerical value and unit symbol.

2.     Line 85. “band” should be “Band”

3.     Figure 1b is black letter, but the others are light gray.

4.     Line 118, 3’Δ1 has consensus.

5.     Figure 2a pA2 is bigger than the others.

6.     Figure 2c and 2d, rectangles in the label should be Δ.

7.     Line 131, remove (D).

Round 2

Reviewer 2 Report

I do not think the authors addressed my concerns. The proposed model is not supported by their results.